# Effect of Parental Migration on the Intellectual and Physical Development of Early School-Aged Children in Rural China

**DOI:** 10.3390/ijerph17010339

**Published:** 2020-01-03

**Authors:** Minmin Li, Ni Zhu, Lingxia Zeng, Duolao Wang, Shaonong Dang, Victoria Watson, Tao Chen, Zhongqiu Hua, Zhaoqing Li, Yijun Kang, Hong Yan, Chao Li

**Affiliations:** 1Department of Epidemiology and Biostatistics, School of Public Health, Xi’an Jiaotong University Health Science Center, Xi’an 710061, China; lmm123@stu.xjtu.edu.cn (M.L.); tjzlx@xjtu.edu.cn (L.Z.); tjdsn@xjtu.edu.cn (S.D.); lizhaoqing@xjtu.edu.cn (Z.L.); tjkyj@xjtu.edu.cn (Y.K.); yh.paper_xjtu@aliyun.com (H.Y.); 2Department of Communicable Disease Control and Prevention, Shaanxi Provincial Center for Disease Control and Prevention, Xi’an 710054, China; zhuni789@163.com; 3Department of Clinical Sciences, Liverpool School of Tropical Medicine Pembroke Place, Liverpool L3 5QA, UK; duolao.wang@lstmed.ac.uk (D.W.); vicky.watson@lstmed.ac.uk (V.W.); taochen@gmail.com (T.C.); 4Department of Nursing, Xi’an Jiaotong University Health Science Center, Xi’an 710061, China; huazhongqiu@gmail.com; 5Nutrition and Food Safety Engineering Research Center of Shaanxi Province, Xi’an 710061, China

**Keywords:** parental migration, intellectual development, physical development

## Abstract

Objective: The purpose of this study is to estimate the effect of parent migration on intellectual and physical development of early school-aged children in rural China. Design: setting and participants: The present cross-sectional study participants were a subset from a controlled, cluster-randomized, double-blind trial. From October 2012 to September 2013, the offspring of women who participated in a large trial were examined in the present study. Wechsler intelligence scale for children (WISC-IV) in which validity and reliability were shown to be satisfactory was used to measure the intellectual function and trained anthropometrists measured weight and height of children using standard procedures. Results: The mean difference of FSIQ scores between non-migration and both-parent migration groups was −3.68 (95%CI: −5.49, −1.87). After adjusting for the confounders, the mean difference of full-scale IQ between non-migration and both-parent migration group was −1.97 (95%CI: −3.92, −0.01), the mean differences of perceptual reasoning index and processing speed index were −2.41 (95%CI: −4.50, −0.31) and −2.39 (95%CI: −4.42, −0.35) between two groups respectively. Conclusion: Our results emphasized the impairment of both-parental migration in intellectual function (FSIQ, PRI, PSI) of children. These findings have important policy implications for the Chinese government to prevent the impairment of left-behind children. Further research is required to clarify the mechanisms by which both-parental migration influence the impairment in intellectual function of children.

## 1. Introduction

Migration is increasing globally, which has resulted in a growing number of children being left behind when their parents migrate [1]. The majority are labour migrants who originate from low-income or middle-income countries [2]. A substantial proportion of rapid economic growth and unprecedented process of urbanization in China are attributed to the largest flow of a massive number of rural labors to urban areas [3,4,5]. More than a third of all children residing in rural China (61 million) are left behind one or both migrant parents [6]. Although migration is tightly linked to labor productivity growth and significantly contribute to rapid economic growth, migration has resulted in millions of children left in their rural communities [4,7]. Whether living in a single-parent migration family or a both-parent migration family, these left-behind children have formed a special youth population in China that deserves serious attention [8].

Previous studies reported the advantages and disadvantages of parental migration for left-behind children. For advantages, migrant workers regularly send remittances to their children left behind, and these additional financial resources enhance the quality of life and socioeconomic status of children left behind [8]. Evidence from the study of Mexico shows households with emigrants are better off than non-migrant households [9]. Results from the study of Philippines shows remittances from emigrants enhance the children’s developmental outcomes in terms of education, physical health and nutrition [10]. For disadvantages, the migration-induced parental absence can cause the injury of child development due to undermined parent-child bonding, lessened family supervision and control, and weakened parental support and guidance [8,11]. In addition, studies conducted in China reported that the parental migration has a negative effect on the high school attendance [12], social anxiety [13] and rank of math scores [4] of rural left-behind children.

Recently, parental migration is predominantly a phenomenon of developing countries. China has been experiencing the history’s largest flow of rural-urban migration in the world [5]. The number of studies which focus on estimating the effect of parent migration on cognitive ability are mostly using exam scores for every school term to estimate the cognitive ability of children [4,14,15,16]. Therefore, the present study aims to estimate the association between parent migration and intellectual development (overall intellectual function and different domains of intellectual function) and physical development of children in rural China.

## 2. Methods

### 2.1. Study Design and Participants

This study is a cross-sectional study to estimate the intellectual and physical development of left-behind children in rural China. The participants were the offspring whose mother took part in a large trial conducted during 2002–2006. The aim of this trial was to determine the effects of prenatal micronutrient supplementation on birth weights, and the details of this trial has been described elsewhere [17,18,19]. Briefly, this double-blinded, cluster-randomized, and controlled trial was conducted in two rural counties. The women in the same villages were randomly assigned to the three daily supplementation groups (multi-micronutrient supplements, folic acid/iron, or folic acid) before recruitment, and women received the same supplement tablets daily from enrollment until delivery. A total of 5828 pregnant women from 531 villages were enrolled in the study, but of these 133 women were lost to follow up, 279 stopped taking supplements and refused to participant and 601 had a spontaneous or induced abortion or other medical condition resulting in fetal loss. There were 4864 births in 4815 women. After excluding the stillbirths and multiple live births, there were 4604 single live births finally.

From October 2012 to September 2013, the primary carer of eligible children was invited to participate in the present study. We excluded children with birth defects and children who had an illness that affected their intellectual development at the time of the study. All eligible children and their parents signed the informed consent. Interviews were administered at the local hospital or school in a standardized manner by trained staff who had demonstrated high levels of consistency with each other in terms of psychological tests and physical measurement of children. We estimated a minimum of 426 children (142 children in each group) would be needed to detect a minimum difference of 5 Full scale intelligence quotient (FSIQ) points that considered clinically significant [20,21] between groups with type 1 error (a = 0.05) and 80% power. Finally, we sampled 1744 children in this study, and this sample size was able to detect the difference of 3.8 FSIQ points between groups.

### 2.2. Anthropometry and Household Information

In the present study, trained anthropometrists measured the weight and height of children using standard procedures. Weight measurement was using an electronic scale (Tanita BC-420, Tanita Corporation, Tokyo, Japan) and recorded to the nearest 0.1 kg. Height (barefooted) was recorded to nearest 0.1 cm using the calibrated stadiometer (Model SZG-210, Shanghai JWFU Medical Apparatus Factory, Shanghai, China).

Interviews were administered at the local school or hospital in a standardized manner by rigorously trained examiner who had demonstrated high levels of consistency with each other in terms of collecting current socio-economic status and household demographics from primary carers, and information repeated years of schooling and type of school was collected from children. Birth outcomes, gestational weeks and others confounders used in present analysis were recorded in the original trial. In detail, the baseline information such as birth day, birth weight (kg), gender, gestation at delivery (weeks) and lower respiratory tract infection (yes, or no) were collected at ≥3 prenatal healthcare visits in different stages of pregnancy. The hospital nursing staff measured birth weight within 1 h of delivery. For home deliveries, township maternal and child health staff visited the homes to measure birth weight and collected birth information within 72 h after delivery. Children age were derived from investigation data subtract birth data. Gestational age at birth was measured as completed days based on the 1st day of the last menstrual period. The purpose of the study was explained to primary carers and children, and written consent from primary carers and child assent were obtained.

### 2.3. Psychological Testing

The latest version of Wechsler Intelligence Scale for Children (WISC-IV) was used to assess intellectual development of children. As one of the most widely used intelligence tests in the world, WISC-IV consists of ten core subtests (Block Design, Similarities, Comprehension, Vocabulary, Picture Concepts, Digit Span, Letter–Number Sequence, Matrix Reasoning, Coding and Symbol Search) and four supplemental subtests (Picture Completion, Information, Cancellation and Arithmetic). The FSIQ, verbal comprehension index (VCI), working memory index (WMI), processing speed index (PSI) and perceptual reasoning index (PRI) could be generated from WISC-IV. In addition, FSIQ represents overall cognitive ability, four other composite scores (VCI, WMI, PSI, PRI) represent different domains of intellectual function [22].

China has standardized the WISC-IV to be culturally appropriate and Chinese norms have been established. Previous studies also reported the validity and reliability of these norms which have been measured and shown to be satisfactory [23]. Currently, the WISC-IV has also been commercialized in China.

### 2.4. Statistical Analysis

All data were double-entered into the data management system and checked manually for completeness. Range, extremum and logical checks were also conducted for accuracy.

In this study, quantitative analysis methods were used. In detail, the distributions of socioeconomic status and household demographic among different parental migration status groups were described by percentages, means and standard deviations. We used multilevel model with township to level 3, village to level 2, and individual to level 1 to compare the weight, height, FSIQ, VCI, WMI, PRI and PSI of children. Statistical significance was set at a *p* value less than 0.05 for all statistical tests, and testing was two-sided. Estimations of the coefficients and 95% confidence intervals were also made in the present study. A household wealth index was constructed with principal component analysis from 16 different household facilities or assets. The household wealth index was categorized into thirds as an indicator of the poorest, middle income and richest households.

Our previous studies identified prenatal and postnatal nutritional status, sex of children, household economic status, schooling of children, educational level of parents and maternal occupation were associated with the intellectual development of children [19,24,25]. In addition, a previous study reported the positive effect of prenatal iron supplementation on intellectual development of children in Nepal [26]. Therefore, we added the following variables as potential confounders: county, the age of children, maternal age, sex of children, educational level of parents, occupation of parents, household wealth index, older siblings of children, child school level, gestational weeks at birth, weight at birth and type of prenatal micronutrient supplementation. Data were analyzed using STATA software, version 12.0.

## 3. Results

Table 1 shows the baseline characteristics of households and children among different parental migration status groups. In summary, the mean age of children and mothers was not different among different parental migration status groups. For non-migration families, the socioeconomic status (education level of parents, household wealth index and schooling of children) of investigated households and children were better than the other groups.

The means of FSIQ was 89.48 (Standard deviation [SD] 13.41), the means of VCI, WMI, PRI and PSI were 87.83 (SD 15.91), 91.05 (SD 12.35), 93.12 (SD 13.90) and 95.68 (SD 13.34), respectively. Children in non-migration family exhibited higher means FSIQ (90.97, SD 13.61), VCI (89.20, SD 16.25), WMI (91.77, SD 12.46), PRI (94.46, SD 13.98), PSI (96.73, SD 13.73), and the mean of height (129.02, SD 6.67) and weight (25.84, SD 4.52) were also slightly higher for children in non-migration family (Table 2).

We used multilevel modelling to estimate the association between parental migration and children’s intellectual and physical development. In univariate analysis, the differences were found in FSIQ (−2.28, 95%CI: −3.79, −0.77), VCI (−1.96, 95%CI: −3.77, −0.16), PRI (−1.71, 95%CI: −3.32, −0.11), and PSI (−2.21, 95%CI: −3.76, −0.67) between the single-parent migration and non-parent migration, while no difference was found in WMI (−0.67, 95%CI: −2.09, 0.75). After adjusting for the confounders, no relevant differences were found in FSIQ (−0.57, 95%CI: −2.08, 0.93), VCI (−1.96, 95%CI: −3.77, −0.16), WMI (−0.67, 95%CI: −2.09, 0.75), PRI (−1.71, 95%CI: −3.32, −0.11), and PSI (−2.21, 95%CI: −3.76, −0.67) between two groups (Table 3).

The mean difference of FSIQ scores between non-migration and both-parent migration groups was −3.68 (95%CI: −5.49, −1.87). After adjusting for the confounders, the mean difference of FSIQ score between two groups was −1.97 (95%CI: −3.92, −0.01). In univariate analysis, the VCI and WMI of children in the non-migration group were significantly higher than the both-parent migration group. The mean differences of VCI and WMI were −2.74 (95%CI: −4.91, −0.58) and −2.18 (95%CI: −3.88, −0.47), respectively. After adjusting for confounders, these mean differences of VCI (mean difference: −0.28, 95%CI: −2.64, 2.07) and WMI (mean difference: −1.25, 95%CI: −3.14, 0.63) between two groups were no longer significant. The mean differences of PRI and PSI between non-migration group and both-parent migration group were significant before and after controlling for confounders. In univariate analysis, the mean differences of PRI and PSI between two groups were −3.82 (95%CI: −5.74, −1.89) and −2.81 (95%CI: −4.66, −0.96). After controlling for the others factors, the mean differences of PRI −2.41 (95%CI: −4.50, −0.31) and PSI −2.39 (95%CI: −4.42, −0.35) between two groups were slightly attenuated and significant (Table 3).

For physical development of children, the weight and height mean differences were not significant not only between non-migration group and both-parent migration group, but also between non-parent migration group and single-parent migration group. The mean difference in weight and height were not significant between non-parent migration and single-parent migration (mean difference: −0.36; 95%CI: −0.81, 0.09; mean difference: −0.12; 95%CI: −0.49, 0.74) or both-parent migration (mean difference: −0.43; 95%CI: −1.02, 0.16; mean difference: 0.29; 95%CI: −0.51, 1.09) group after adjusting for confounders (Table 3).

## 4. Discussion

### 4.1. Main Finding of This Study

The major finding in this study is the negative effect of both-parent migration on intellectual development of children. Specifically, children in both-parent migration group displayed lower FSIQ, which represents overall cognitive ability, and two other composite scores (PSI, PRI). However, no effect of single-parent migration vs. non-parent migration were found on the intellectual development of early school-aged children in rural China. For physical development, no difference was found among different parent migration status groups.

### 4.2. What Is Already Known on This Topic

With the acceleration of urbanization and industrialization over the past four decades, china experienced the largest rural-urban migration in human history. The problems associated with separating children from parent duo to transient are inevitable. A considerable amount of research was published on the effects of parental migration on left behind children. These studies demonstrated that children with parent migration in rural China experience more challenges compared to their counterparts living with parents with respect to loneliness, depression, anxiety, nutrition, and behavioral problem. However, the number of studies regarding cognitive ability especially intellectual development of left behind children in rural China was limited. Moreover, the method used to measure cognitive ability of children in the other studies was mostly the final exam scores for every school term, and these exams were designed by each school to estimate mastery of academic subjects covered in school term. Therefore, the exam scores are not directly comparable, and not nearly enough to correctly reflect cognitive ability of children.

For cognitive ability of children, results from previous studies were inconsistent. A Chinese study reported the negative effect of parental migration on children’s school performance, and by having a migratory parent would reduce a child’s math score rank by 15.6% [4]. Another Chinese study reported the negative impacts of both-parent migration on children’s cognitive development, reducing examination scores by 5.1 percentile points for Chinese and 5.4 percentile points for math [16]. In contrast, evidence from a Chinese study showed there was no effect of parental migration on cognitive ability and physical development of children [27].

For the physical development of children, results from previous studies were also inconsistent. Results from a Chinese study showed that male left-behind children were at higher risk of being overweight [28]. Another study conducted in Sweden reported that children of immigrants had higher intake of sucrose. Furthermore, these children had a higher risk of having low physical activity and being overweight [29]. In contrast, results from a study conducted in Chinese rural areas showed that there were no significant differences in weight and height between left-behind children and control group [30].

### 4.3. What This Study Adds

In the present study, we found no effect of parent migration on physical development of children. But negative effect of parent migration on cognitive ability was found in the present study. Compared to the other studies, the method used to measure the intellectual development of children in present study was WISC-IV which can better evaluate the full-scale and different dimensions of intellectual development of children. Results from another study showed no effect of parent migration on cognitive ability, which is inconsistent with the present study. It is possible that the rate of both-parents migration was less than 30 percent in that sample and the majority of children were still living in single-parent migration family (around 60 percent was a mother who could provide amount of caring and parenting) [27]. The observed association between parent migration and impairment of intellectual development may be attributable to several mechanisms. On the one hand, most guardians of left-behind children can not give deliberate attending and scientific management because of old age, limited education level and valetudinarianism [30]. On the other hand, the mother who is working out of her home may negatively influence the breastfeeding and micronutrients supplementation of left-behind children [30]. Results from a large randomized trial showed that exclusive and prolonged breastfeeding improves children’s cognitive development [31]. In addition, micronutrients also plays an important role in children’s intellectual development. For example, a cohort follow-up study conducted in Nepal reported iron supplementation were positively associated with children’s motor function [26]. These results could provide indirect evidence supporting the possibility that emotional, behavioral problems may be the potential mechanism and explanation of these findings.

In this study, although the differences of weight and height among different parental migration status groups were not significant, the mean values of weight and height were higher in non-migration group. This finding might be explained by the economically non-migrant households were better than migrant households. Although previous studies reported migrant households were better by regularly sending remittances to left behind family members [8,9,10]. In contrast, we found that economically non-migrant households were better than migrant households. In the present study, in order to truly reflect the environment of children, the economic situation of primary carers for migration household was estimated. In addition, the children investigated in the present study were in the early school ages (7–10 years), and previous studies reported that migrant parents tended to accumulate money for their children’s education [32]. Therefore, there will not be much remittances sent to the left behind family members in the present study. With the rapid urbanization, coal mining was rapidly developed within the present study field, and many non-migration households have benefitted from it accordingly in recent years.

Results from the present study identified the negative effect of parental migration especially both-parents migration on intellectual development of children but not on the physical development. Findings from previous studies indicated the importance of intervention from different levels in the healthy development of children. From the individual level, the prevention program will improve left-behind children’s coping skills with the separation from parents as well as emotional and subsequent social problems. From the family level, migrant parents should focus on more frequent parent-child, long-distance communication and children’s emotional needs. From the community level, improving the teacher-child relationship and economic situation of left-behind children are needed [33].

### 4.4. Strengths and Limitations of This Study

Several limitations and strengths of the present study should be pointed out. Firstly, we did not include all possible confounders in our analysis. In detail, parental intellectual function has been reported as a strong factor of children’s intellectual function [34], and it may also affect development of children during infancy via feeding practice and physical activity. The data of maternal psychological distress that may be associated with parenting behavior was also not collected. However, a large birth cohort study demonstrated a similar result after the adjustment for maternal depression and anxiety, when analyzing the association between behavioral problems and birth weight [35]. Secondly, we did not collect the data about the starting time and duration of parental migration. Therefore, it is impossible for us to determine the causation. The original cohort study was designed to estimate the long-term effects of prenatal and postnatal nutritional status on further intellectual and physical development of children [36]. Within the limitations our results also provide evidence for the negative effect of parental migration on healthy development of left-behind children. Finally, this study was an information-seeking study, further intervention or qualitative studies are needed to provide solution to improve intellectual development of left-behind children. Despite these limitations, the very common phenomenon of left-behind children will continue in China for the foreseeable future and the negative effects of parental migration on children will also persist [37]. Our findings contribute to the literature and have important implications to the local communities for the development of comprehensive quality of life-behind children in rural China.

The present study also has several strengths. Due to the nature of longitudinal data, gestational information such as gestational weeks at birth and birth weight was collected at the time of birth, ensuring the analysis did not suffer the potential bias that may be introduced when using the recalled information. In addition, we used a standardized measure of intellectual functioning available for use in different cultures background and the latest version of WISC which proved to be satisfactory in validity and reliability in Chinese.

## 5. Conclusions

In conclusion, our results emphasized the impairment of both-parental migration in intellectual function (FSIQ, PRI, PSI) of children. But the effect of single-parental migration on intellectual development of children was not found. In addition, impairment of both or single parental migration in physical development was not evident. The strong and clear association between parental migration and intellectual development of children in this study highlight the importance of conducting intervention to target concerns in left-behind children whose parents were both left. These findings have important policy implications for the Chinese government to prevent the impairment of left-behind children. Further research is required to clarify the mechanisms by which both-parental migration influence the impairment in intellectual function of children.

## Figures and Tables

**Table 1 ijerph-17-00339-t001:** Baseline Characteristics of Children, Their Mothers and Households in Different Group ^a^.

	Parental Migration Status		
	Both-Parent Migration	Single-Parent Migration	Non-Migration
Number of children	249	401	1090
Children age			
Mean (SD)	8.75 ± 0.80	8.77 ± 0.81	8.79 ± 0.84
Gender			
Boy	140 (56.2%)	243 (60.6%)	659 (60.5%)
Girl	109 (43.8%)	158 (39.4%)	431 (39.5%)
Maternal age (years)			
Mean (SD)	32.10 ± 3.91	34.53 ± 4.48	34.18 ± 4.55
Women’s education			
Primary	82 (32.9%)	170 (42.5%)	381 (35.1%)
Secondary	157 (63.1%)	213 (53.3%)	536 (49.3%)
High school+	10 (4.0%)	17 (4.2%)	170 (15.6%)
Father’s education			
Primary	25 (10.1%)	76 (18.9%)	187 (17.2%)
Secondary	185 (74.6%)	279 (69.6%)	630 (58.0%)
High school+	38 (15.3%)	46 (11.5%)	269 (24.8%)
Women’s occupation			
Farmer	73 (31.7%)	335 (84.4%)	739 (68.6%)
Others	157 (68.3%)	62 (15.6%)	339 (31.4%)
Father’s occupation			
Farmer	59 (25.7%)	139 (34.7%)	440 (40.9%)
Others	171 (74.3%)	262 (62.4%)	635 (59.1%)
Household wealth index			
Poorest—1st tertile	103 (41.4%)	174 (43.4%)	294 (27.0%)
2nd tertile	99 (39.7%)	159 (39.7%)	324 (29.7%)
Richest—3rd tertile	47 (18.9%)	68 (16.9%)	472 (43.3%)
Parity			
0	192 (77.1%)	233 (58.1%)	670 (61.5%)
1	53 (21.3%)	142 (35.4%)	358 (32.8%)
≥2	4 (1.6%)	26 (6.5%)	62 (5.7%)
School			
Village	112 (45.0%)	179 (44.6%)	305 (28.0%)
Township	106 (42.6%)	183 (45.7%)	430 (39.5%)
County and plus	31 (12.4%)	39 (9.7%)	355 (32.5%)
Birth weight			
Mean (SD)	3.17 ± 0.41	3.22 ± 0.46	3.20 ± 0.43
Gestation at delivery (weeks)			
Mean (SD)	39.88 ± 1.74	39.83 ± 1.74	39.86 ± 1.63
Lower respiratory tract infection			
Yes	146 (63.5%)	257 (64.6%)	749 (69.7%)
No	84 (36.5%)	141 (35.4%)	325 (30.3%)
Treatment group			
Folic acid	89 (35.7%)	126 (31.4%)	388 (35.6%)
Folic acid/Iron	62 (24.9%)	134 (33.4%)	366 (33.6%)
Multi micronutrients	98 (39.4%)	141 (35.2%)	336 (30.8%)

^a^ Values are given as Mean ± SD or the percentage of the study population in different group.

**Table 2 ijerph-17-00339-t002:** Mean Scores on WISC-IV Test by migrant of parents ^a^.

	Both-Parent Migration	Single-Parent Migration	Non-Migration	Total
Intellectual development				
FSIQ	86.25 ± 12.45	87.44 ± 12.83	90.97 ± 13.61	89.48 ± 13.41
VCI	85.16 ± 14.45	85.75 ± 15.43	89.20 ± 16.25	87.83 ± 15.91
WMI	89.16 ± 12.03	90.26 ± 12.12	91.77 ± 12.46	91.05 ± 12.35
PRI	89.54 ± 13.07	91.70 ± 13.67	94.46 ± 13.98	93.12 ± 13.90
PSI	93.82 ± 12.54	93.97 ± 12.48	96.73 ± 13.73	95.68 ± 13.34
Physical development				
Weight, kg	25.38 ± 4.36	25.11 ± 4.17	25.84 ± 4.52	25.59 ± 4.43
Height, cm	128.52 ± 6.59	128.32 ± 6.46	129.02 ± 6.67	128.79 ± 6.61

Abbreviation: FSIQ, Full-scale Intelligence Quotient. VCI, Verbal Comprehension Index. WMI, Working Memory Index. PRI, Perceptual Reasoning Index. PSI, Processing Speed Index. ^a^ Values are given as Mean ± SD.

**Table 3 ijerph-17-00339-t003:** Comparison of WISC-IV Test Scores by parental migration status ^a^.

	Single-Parent Migration		Both-Migration	
	Unadjusted Analysis	Adjusted Analysis ^b^	Unadjusted Analysis	Adjusted Analysis ^b^
	MD (95%CI)	MD (95%CI)	MD (95%CI)	MD (95%CI)
Intellectual development				
FSIQ	−2.28 (−3.79, −0.77)	−0.57 (−2.08, 0.93)	−3.68 (−5.49, −1.87)	−1.97 (−3.92, −0.01)
VCI	−1.96 (−3.77, −0.16)	−0.46 (−2.26, 1.35)	−2.74 (−4.91, −0.58)	−0.28 (−2.64, 2.07)
WMI	−0.67 (−2.09, 0.75)	0.35 (−1.10, 1.80)	−2.18 (−3.88, −0.47)	−1.25 (−3.14, 0.63)
PRI	−1.71 (−3.32, −0.11)	−0.14 (−1.75, 1.47)	−3.82 (−5.74, −1.89)	−2.41 (−4.50, −0.31)
PSI	−2.21 (−3.76, −0.67)	−1.03 (−2.59, 0.53)	−2.81 (−4.66, −0.96)	−2.39 (−4.42, −0.35)
Physical development				
Weight, kg	−0.16 (−0.86, 0.54)	0.07 (−0.58, 0.71)	0.28 (−0.34, 0.90)	0.43 (−0.16, 1.02)
Height, cm	−0.47 (−1.50, 0.57)	−0.17 (−1.05, 0.71)	−0.19 (−1.11, 0.73)	−0.29 (−1.09, 0.51)

Abbreviation: MD, Mean difference. CI, Confidence interval. FSIQ, Full-scale Intelligence Quotient. VCI, Verbal Comprehension Index. WMI, Working Memory Index. PRI, Perceptual Reasoning Index. PSI, Processing Speed Index. ^a^ Multilevel models were used to assess the association between prenatal micronutrient supplement and IQ in early school aged children, with township to level 3, village to level 2, and individual to level 1. ^b^
*p* value adjusted for variables of county, the age of children, maternal age, sex of children, educational level of parents, occupation of parents, household wealth index, older siblings of children, child school level, gestational weeks at birth, weight at birth, type of prenatal micronutrient supplementation.

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
