# Peer review of "Effect of Parental Migration on the Intellectual and Physical Development of Early School-Aged Children in Rural China"

_ijerph, 2020, doi:10.3390/ijerph17010339_

Round 1

Reviewer 1 Report

I am of the view there is a need to proof read the article, e.g. 66-67; 85. That is the common mistake that I have noted. I also feel like the results, discussion and conclusion sections can be expanded. On methodology, the section that talks about interviewing parents I think is in the manner the approach was used, unless the type of interviewing is clearly described. That section brings an aspect of qualitative nature which this research does not employ. The conclusion sounds weaker and needs further strengthening. Overall, this is a nice piece of research, offering interesting insights on the impact of migration on children left behind.

Author Response

Dear reviewer:

  Thanks for your comments and suggestions on this manuscript. It’s a great honor to arrange our manuscript reviewed by you.

Next, we will provide a point-by-point response to your comments.

Point 1: I am of the view there is a need to proof read the article, e.g. 66-67;85. That is the common mistake that I have noted.

Response 1: Your comment is very valuable to us. It’s really due to our carelessness. We have revised the mistakes with red font and checked the full text to avoid similar mistakes in other places.

Point 2: I also feel like the results, discussion and conclusion sections can be expanded.

Response 2: The results in this manuscript have been reanalyzed  (table 3). In this section, we compared the difference of intellectual and physical development not only between non-parent migration group and single-parent migration group, but also between non-parent migration group and both-parent migration group. Meanwhile, we described the results in detail (e.g. 179-210) and expanded the discussion (e.g. 216-234) and conclusion (e.g. 329-339) according to the results. See red font in the manuscript. We hope our revision could be convincing.

Point 3: On methodology, the section that talks about interviewing parents I think is in the manner the approach was used unless the type of interviewing is described. That section brings an aspect of qualitative nature which this research does not employ.

Response 3: After watching your comments, we went through the method section again and found that we did not describe the interview clearly. We have revised this section e.g. 107-122.

Point 4: The conclusion sounds weaker and needs further strengthening.

Response 4: The conclusion had been expanded and strengthened, e.g. 35-39 and e.g.329-339.

The revised manuscript can see the attachment.

Reviewer 2 Report

This seems to be part of a larger study deisgned for other purposes and if so more details of this study should be noted

There also should be some discussion of the significance of these results

Author Response

Dear reviewer:

  Thanks for your comments and suggestions on this manuscript. It’s a great honor to arrange our manuscript reviewed by you.

Next, we will provide a point-by-point response to your comments.

Point 1: This seems to be part of a large study designed for other purposes and if so more details of this study should be noted.

Response 1: You are right, the participants in the present study were from a large cluster-randomized, double-blind, controlled trial, which examined the effects of antenatal micronutrient supplementations on perinatal outcomes (www.isrctn.com: ISRCTN08850194). From 2012 to 2013, we followed up all single live births whose mothers participated in the antenatal trial to explore the intellectual and physical development of those children. The more details of this study have described in the revised manuscript.

Point 2: There also should be some discussion for the significance of these results.

Response 2: We described the significance of the results in detail e.g. 179-187 and e.g.206-210 and expanded the discussion e.g.221-234 and conclusion e.g.329-339according to the results. See red font in the manuscript. We hope our revision could be convincing.

The revised manuscript can see the attachment
